# Availability, price and affordability of insulin, delivery devices and self-monitoring blood glucose devices in Indonesia

**Hesty Utami Ramadaniati**[1]*, **Yusi Anggriani**[1], **Molly Lepeska**[2], **David Beran**[3], **Margaret Ewen**[2]

**1** Faculty of Pharmacy, Universitas Pancasila, South Jakarta, Indonesia, **2** Health Action International, Amsterdam, The Netherlands, **3** Division of Tropical and Humanitarian Medicine, University of Geneva, Geneva, Switzerland

* hesty.utami@univpancasila.ac.id

**Data Availability Statement:** All relevant data are within the manuscript and its Supporting Information files

## Abstract

Insulin is essential for the survival of people with type 1 diabetes and for better management of people with type 2 diabetes. People with diabetes using insulin also require self-monitoring blood glucose (SMBG) devices (e.g., meters, strips, continuous monitoring systems) for day-to-day management. It is essential to ensure that insulin and these devices are available and affordable. This study aimed to evaluate the availability, price, and affordability of insulin and SMBG devices in Indonesia using an adaptation of the World Health Organization/Health Action International (WHO/HAI) price survey. A total of 34 public health facilities (hospitals, primary healthcare centres/Puskesmas) and 37 private pharmacies were sampled. Information from three major online marketplaces was also collected. Prices were expressed as median patient prices (US$). Affordability was defined as the number of days' wages needed by the lowest paid unskilled government worker (LPGW) to purchase 30 days' supply of insulin, delivery devices and SMBGs. Availability of analogue insulin was slightly higher in public facilities (63.6%) than in the private sector (43.2%), with no human insulin available in both sectors. Conversely, better availability was observed in private facilities for SMBG devices as public sector facilities did not supply devices for self-testing. Median prices for 1000IU analogues varied between the public sector (US$ 5.26) and the private sector (US$11.24). The highest median price of analogues was seen in online marketplaces (US$ 28.65). The least costly median price of SMBG devices were observed in online platforms (meter: US$ 18.37, test strip: US$ 0.27, lancet: US$ 0.02). A low-income person had to work 2–3 days to buy 1000IU of analogues. It required 5–7 days' and 4–5 day's wages to purchase a meter and a month's supply of test strips, respectively. The availability and affordability of insulin and SMBG devices remain important issues in Indonesia requiring holistic approaches for further improvement.

**Funding:** Health Action International. The funders had no role in study design, data collection and analysis, decision to publish, or preparation of the manuscript.

**Competing interests:** The authors have declared that no competing interests exist

## Introduction

Diabetes remains one of the growing major global public health concerns. The International Diabetes Federation reported approximately 537 million people across the globe were living with diabetes in 2020 and this number is predicted to increase to more than 700 million by 2045. They also estimate that over 80% of people with diabetes live in low- and middle-income countries (LMICs), thus imposing a significant burden on these already poor-resourced countries [1].

Poorly controlled diabetes results in increased morbidity and mortality. In 2020, diabetes resulted in 6.7 million deaths globally among people aged 20–79 years, and it accounted for 32.6% of deaths in people under 60 years [1]. Diabetes also has a financial impact on people and healthcare systems [2]. Total global health expenditure for managing diabetes in adults amounted to US$ 966 billion in 2021 [1]. Meanwhile, a systematic review of direct medical costs for diabetes in LMICs found a high degree of cost variability with annual inpatient costs between less than US$ 20 and US$ 100, and medicine-related costs ranging from less than US$ 20 per year to over US$ 500. Further, the annual cost of insulin for a person with type 1 diabetes varied between US$ 60 to more than US$ 300. Similar variations were observed in type 2 diabetes with insulin costs of around US$ 50 –US$ 100 for each patient annually [3]. The costs of managing complications might be much higher. A study estimating the direct medical costs of type 2 diabetes and its complications in Indonesia revealed the total cost of treatment was US$ 576 million in 2016 with nearly three-quarters being used to treat complications. The study highlighted the importance of early diagnosis and optimal management of complications as an effective approach for cost-saving [4].

Despite the centenary of the discovery of insulin, its availability and affordability remain a protracted challenge in many parts of the world [5]. Globally, 9 million people with type 1 diabetes and 63 million people with type 2 diabetes require insulin [6]. Access to affordable insulin is crucial, yet access to self-monitoring blood glucose (SMBG) devices (e.g., meters, strips, lancets, continuous monitoring systems) is also pivotal for diabetes management. SMBG devices are universally considered to be an integral part of type 1 diabetes management and insulin-treated type 2 diabetes for optimizing the safety and efficacy of insulin regimens [7]. Unfortunately, poor availability and high costs of insulin, delivery devices and SMBG devices compromises treatment, leading to suboptimal outcomes and the emergence of complications [8]. During the 75[th] World Health Assembly, World Health Organization (WHO) member states set five new global targets aimed to be achieved by 2030 as part of recommendations for strengthening and monitoring diabetes. One of the targets is 100% access to affordable insulin and SMBG devices for people living with type 1 diabetes. This ambitious target signifies the importance of accessible and affordable SMBG devices in the management of type 1 diabetes [9].

Indonesia is the largest archipelagic country comprising over 17,000 islands stretching along the equator in Southeast Asia. Indonesia is a middle-income country with health expenditure at 3.41% of gross domestic product in 2020, which is relatively low compared to its neighboring countries such as Malaysia, Singapore and Thailand [10]. In 2014, the government implemented a mandatory national health insurance/NHI scheme (called *Jaminan Kesehatan Nasional*/JKN). As of September 2023, JKN covered nearly 95% of population (262.8 million members), making JKN the largest single-payer system in the world [11]. JKN patients receive medicines at no cost at JKN-affiliating health facilities. Medical devices are provided free-of-charge for JKN in-patients (part of case-based reimbursement payment), but out-patients must pay out-of-pocket (OOP) for the devices. It is important to note that more than half of health spending in Indonesia is sourced from the government, yet the percentage of

OOP health expenditure is relatively high at 31.8% with a significant proportion being on medicines [12].

Indonesia still heavily relies on imported medicines, raw materials and medical devices. Total imports of medical devices reached US$ 2,183 million in 2020 (accounting for more than 80% of the medical devices market), increasing from US$ 1,550 million in the preceding year [13]. This situation may influence access to insulin and SMBG devices in Indonesia with the increasing prevalence of diabetes. The 2018 report of the Indonesian Basic Health Survey -a national survey every five years- showed the national prevalence of diabetes based on a self-report diabetes diagnosis approach (population aged $\geq$ 15 years old) was 2.0% [14], a rise of 0.5% since the 2013 survey [15].

Insulin requires a doctor's prescription in Indonesia. It is available in public and private health facilities. For public facilities, insulin is only available in secondary and tertiary health care facilities. Government-owned primary healthcare facilities (i.e. *Pusat Kesehatan Masyarakat*/Puskesmas) are not authorized to procure and dispense insulin. Generally, insulin is provided free in the public hospitals and private hospitals affiliated with JKN. However, public hospitals also provide insulin for non-JKN patients covered by private insurance or to people paying OOP. SMBG devices are available in nearly all public facilities for in-facility testing but not supplied for home-use. Insulin and SMBG devices are readily available in private medicine outlets (pharmacies, clinics, hospitals) for in-facility testing and home monitoring. As with private outlets, insulin and SMBG devices are accessible through online marketplaces despite regulations stating that medicines and medical devices should be procured, distributed and dispensed by authorized parties (health facilities, licensed distributors). In terms of licenses to sell medical products, there are two types of online marketplaces: regulated and unregulated. Regulated online marketplaces are defined as digital platforms approved by Indonesia's Ministry of Health for selling medicines [16]. Seven platforms were approved at the time of this study; the other platforms were unregulated. However, some unregulated platforms collaborated with other parties to verify the licenses of the online stores of physical pharmacies, providing them a special logo and categorizing them as verified sellers of medical products.

To the best of our knowledge, little is known on access to SMBG devices. Previous studies mainly focused on insulin. Therefore, this study aimed primarily to assess the availability, prices, and affordability of insulin and SMBG devices in Indonesia. Secondary objectives were to compare the cost of monthly glucose self-testing to the cost of treatment with insulin, and to put forward some measures to improve access to insulin and SMBG devices in LMICs.

## Methods

### Study setting and sampling

A cross-sectional facility-based survey was conducted using an adaptation of the World Health Organization/Health Action International (WHO/HAI) price survey [17] plus data was collected from online marketplaces. We collected availability and price data from facilities in 4 provinces (Jakarta, North Sumatra, West Kalimantan and East Nusa Tenggara). Jakarta, the capital of Indonesia, is located on Java Island, whilst the other three provinces are spread on different islands in the Western, Central and Eastern parts of Indonesia. Three sectors were surveyed: public (hospitals, Puskesmas), private pharmacies and online marketplaces. For each province, data were collected in two areas: the capital and a district in a rural area. In each area, the major public hospitals and/or Puskesmas were sampled, plus five private retail pharmacies (randomly chosen from those within 5 km of the public hospitals). In total, we collected data from 34 public facilities (12 hospitals, 22 Puskesmas) and 37 private pharmacies. For online marketplaces, data was collected from three platforms (one unregulated and two regulated).

## Survey medicine and SMBG devices

Data were collected for insulin (analogue and human insulin), insulin delivery devices (pen needles, insulin syringes) and SMBG devices (meters, test strips, lancets and continuous glucose monitoring systems/CGMs) found in the outlets and online marketplaces. We excluded multifunction meters and strips.

## Data collection and analysis

Surveying medicine outlets was conducted in September–October 2022, whilst online marketplace survey was done in October 2022 –January 2023. Availability was defined as the percentage of outlets that stocked insulin, delivery devices and SMBGs devices on the day of data collection. Mean availability per sector was calculated (excluding the marketplaces) and the number of SMBG brands found per outlet. Availability was classified as follows: <30% very low, 30%-49% low, 50%-80% fairly high, >80% very high [18]. We excluded availability data of insulin and SMBG devices from Puskesmas. Insulins are supplied to people in secondary and tertiary facilities (i.e., hospitals) but not Puskesmas. In addition, Puskemas stock meters and strips for in-facility testing only (not for sale for self-testing at home). For the SMBG brand analysis, we made no distinction between different models of a brand.

Retail prices were calculated in US dollar by converting the Indonesian currency using the exchange rate on OANDA (https://www.oanda.com/currency/converter) on the first day of data collection (US$ 1 = IDR 14864.31). We calculated median, minimum and maximum unit prices per sector. For insulin, unit price was defined as price to purchase 10 ml (1000IU) of insulin. For online marketplace, delivery costs were not included in the retail prices. The WHO/HAI methodology reports median prices in local currency and as a median price ratio. In this study, we reported the unit price in US$ only.

Affordability was expressed as the number of days required by the lowest paid unskilled government employee to purchase insulin (1000IU), one meter, a pack of 50 test strips or two CGM sensors at median prices [5]. The minimum daily wage was approximately US$ 3.50 [19].

One month's supply was applied in the comparison of SMBG costs for glucose self-testing versus treatment costs (i.e., analogue insulin and pen needles). The supply was defined as follows:

- Meter: 1/24th of the median unit price (assumption: a meter would last two years)

- Test strips: 150 (Note: The Indonesian Association of Endocrinology does not explicitly recommend daily testing levels except several times a day for uncontrolled people, including before and following each meal and at bedtime. This is similar to the American Diabetes Association recommendation of 6–10 tests per day for people with type 1 diabetes) [20]

- Insulin: 600IU rapid-acting and 600IU long-acting analogues (Defined Daily Dose of 40IU/day) [21]

- Pen needles: 60 (one needle per day for each analogue insulin)

- Lancets: 8

We used the WHO/HAI threshold of spending not more than 1 day's wage to purchase 30 day's supply (insulin or SMBG devices) to define affordable [17].

## Ethics statement

This study was approved by the Ethics Committee of the Faculty of Medicine University of Indonesia–Cipto Mangunkusumo Hospital (protocol number: 22-08-0893). Written

informed consent was obtained from the respondents (pharmacists or pharmacist assistants) prior to the survey.

## Results

### Availability and commonly encountered brands

**Insulin and delivery devices (syringes and pen needles).** As depicted in Fig 1, insulins were available in more than 60% of the public hospitals (N = 14/22, 63.6%). It is important to note that none of the public facilities provided human insulin as they only procured analogue insulins. Rapid-acting analogues were the most frequently found insulin in the public sector. In private community pharmacies, insulins were available in less than half of the study outlets (n = 16/37, 43.2%). As with the public sector, only analogues were found in the private sector. Long-acting and rapid-acting analogues the commonest analogues found in the private pharmacies. When comparing the availability of any analogues across regions, availability in public facilities was fairly high (more than 50%) in three provinces but not in East Nusa Tenggara where availability was 33.3%. In private pharmacies, availability of any analogue was variable ranging from 36.4% in East Nusa Tenggara to 66.7% in North Sumatera.

As human insulins (vials) were not available in public health facilities, it is unsurprising that insulin syringes were not found in this sector. Low availability (2.7%) of insulin syringes was found in private pharmacies despite zero availability of human insulin. The availability of pen needles, for use with analogue pre-filled pens, was less than half in public (44.1%) and private facilities 48.6% (see Fig 1). Regional variations were documented in the availability of pen needles. In the public sector, the highest availability was in Jakarta (66.7%) with variable availability in the other three provinces ranging from 0% (West Kalimantan) to 50.0% (East Nusa Tenggara). Availability of pen needles was also variable in the private sector, ranging from 16.7% (Jakarta) to 63.6% (North Sumatera).

**Meters.** Availability of meters for self-testing was much higher in private pharmacies (45.9%) than in the public sector 0% (Fig 2). Variation was observed across the survey provinces where 100% availability of blood glucose meters for in-facility use was found in public hospitals in Jakarta and none provided glucose meters (for purchase or in-facility testing) in

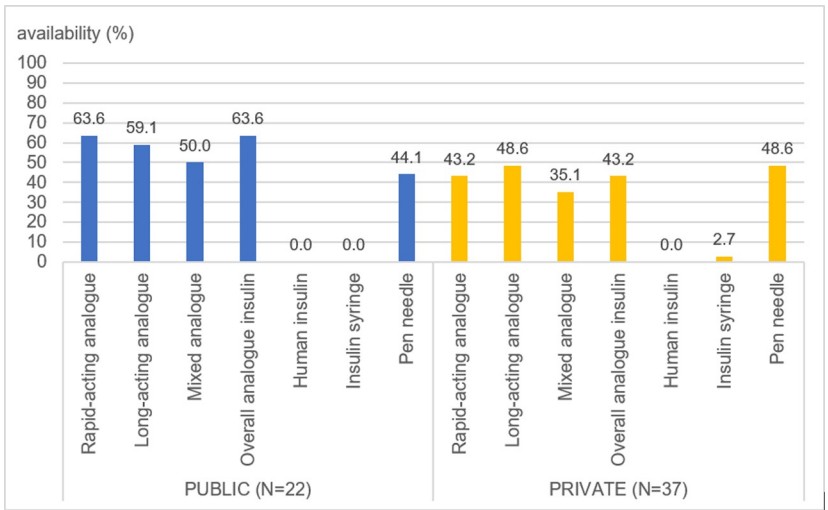

**Fig 1. Availability (in percentage) of insulin and insulin delivery devices (insulin syringe, pen needle) by sectors.**

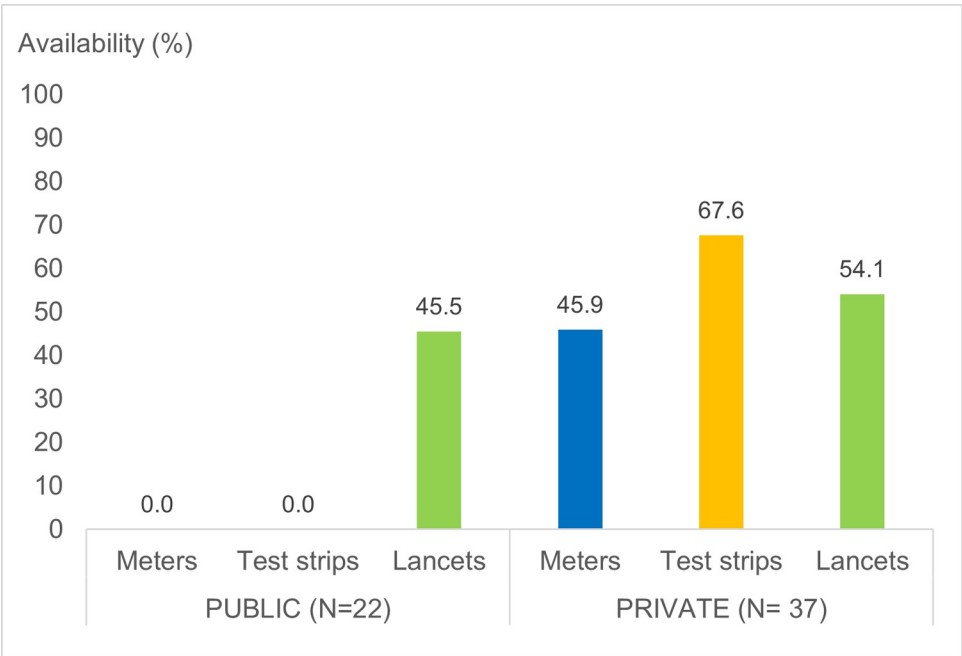

**Fig 2. Availability (in percentage) of self-monitoring blood glucose devices by sector.**

the other three provinces. In the private sector, regional availability of meters for self-testing was also variable, ranging from 9.1% in West Kalimantan to 77.8% in North Sumatra.

**Test strips.** As shown in Fig 2, a similar trend was seen in test strips for self-testing. Availability in the private sector (67.6%) was considerably higher than in the public sector (0%). As with meters, we observed variation across the study provinces. In the public sector, test strips for in-facility glucose testing were only found in Jakarta and none provided test strips for self-testing, whilst in private pharmacies the availability (either for purchase or in-facility testing) was considered good ranging from 50% in Jakarta to nearly 90% in North Sumatra.

**Lancets.** There is little difference in availability of lancets for self-testing between the public (45.5%) and private sectors (54.1%).

**CGMs.** None of the study outlets provided CGMs as no brands are registered in Indonesia. However, one CGM brand (i.e., Free-style Libre® by Abbott) could be easily found in online marketplaces, especially the unregulated platform.

## Brands of SMBG devices (meters, test strips)

Table 1 details the number of brands of meters and test strips found in the private sector and online marketplace. The public sector (government-owned hospitals and Puskesmas) is excluded as devices for self-testing were unavailable. Overall, most private pharmacies provided only one brand of meter or test strip. EasyTouch® was the commonest brand of meter and test strip found in the outlets across the provinces. In the online marketplaces, 340 online shops were selling meters and tests strips across the three digital platforms. The majority of the shops provided one brand of meter (70%) and one brand of test strip (89%) with the remainder selling two or more brands. Accu-Chek® was the most frequent meter sold online, whilst the most sold test strip was GES®.

**Table 1. Percentage of brands found and leading brands by sector and province.**

| Sector | Province | Meters | | | | Test Strips | | | |
|---|---|---|---|---|---|---|---|---|---|
| | | Number of brands found per outlet* | | | Most found brand | Number of brands found per outlet* | | | Most found brand |
| | | 1 | 2–5 | ≥6 | | 1 | 2–5 | ≥6 | |
| **Private** | Jakarta (N = 6) | 50.0% | 0% | 0% | EasyTouch® Benecheck® Gluco Dr® | 50.0% | 0% | 0% | EasyTouch® Benecheck® Gluco Dr® |
| | North Sumatra (N = 9) | 55.6% | 22.2% | 0% | EasyTouch® | 55.6% | 33.3% | 0% | EasyTouch® |
| | West Kalimantan (N = 11) | 0% | 9.1% | 0% | Accu-Chek® Gluco Dr® | 36.4% | 18.2% | 0% | Accu-Chek® |
| | East Nusa Tenggara (N = 11) | 18.2% | 45.5% | 0% | Gluco Dr® Autocheck® | 27.3% | 45.5% | 0% | Autocheck® |
| | Overall (N = 37) | 24.3% | 21.6% | 0% | EasyTouch® | 40.5% | 27.0% | 0% | EasyTouch® |
| **Online Marketplace** | | 70% | 29% | 1% | Accu-Chek® | 89% | 10% | 1% | GES® |

Note: Public sector data excluded as availability of meters and strips was 0%

## Retail prices

**Insulin and delivery devices (syringes and pen needles).** Fig 3 gives the median retail prices of 1000IU human and analogue insulins. For analogues, median prices per 1000IU were US$ 5.26 in the public sector and US$ 11.24 in private pharmacies. Median prices of analogues in the online marketplaces were higher at US$ 28.65. The greatest difference between the minimum and maximum price was found in the public sector and online marketplace, with the highest priced analogue nearly 14-times the price of the lowest priced analogue. The price differences in private outlets were approximately 9-fold. Human insulin was found only in online platforms, with a median price of US$ 7.65 (range: US$ 6.7–38.24). With respect to insulin delivery, the median unit price of insulin syringes and pen needles was US$ 0.15 and US$0.25, respectively.

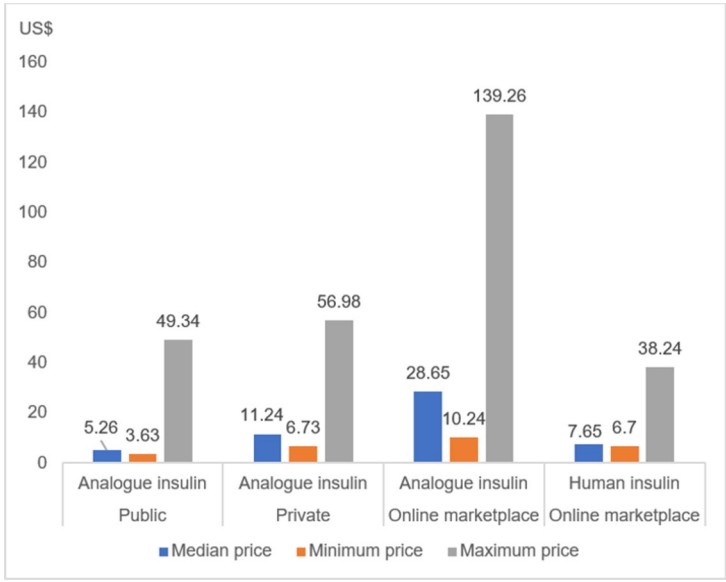

**Fig 3. Median, minimum and maximum prices of insulin by sector.**

**Meters.** As illustrated in Table 2, the median price per meter was slightly lower in online marketplaces (US$ 18.37) than in private pharmacies (US$ 23.9). However, massive price variation was observed when purchasing from online marketplaces, with price differences exceeding US$ 100 a meter.

**Test strips.** The median price per strip in online marketplaces (US$ 0.27) was lower than in private pharmacies (US$ 0.34). There were large differences between minimum and maximum prices in private pharmacies and online marketplaces (see Table 2). People could spend eight times and ten times more to purchase the highest priced strips in private pharmacies and digital platforms, respectively.

**Lancets.** The median unit price of lancets was highest in public facilities (US$ 0.05) with the identical median price in both private pharmacies and online marketplace (US$ 0.02). The difference between the highest-priced and lowest-priced lancets in all sectors was large. Patients purchasing the highest priced lancets spent at least ten times more than the lowest priced products in private and public outlets. The highest priced lancets were approximately 275 higher than that of the cheapest ones in the online markets.

**CGMs.** The median prices of a CGM reader and a sensor, when purchased from the online marketplace, were similar at approximately US$ 106 per unit. The price difference between the highest and lowest readers and sensors ranged around 4–5 times.

## Affordability

Based on median prices, the lowest paid unskilled government employee paying out-of-pocket would have to work 1.5 days or 3.2 days to purchase 1000IU/mL analogue insulin in the public or private outlets, respectively. Analogue insulin procured through online marketplaces was less affordable requiring approximately eight days' wages, however, human insulin required only two days' wages (albeit there were few sellers of human insulin). As described in Table 3, it required 6.8 and 5.3 days' wages to purchase one meter in private pharmacies and online, respectively. To purchase a month's supply of test strips required approximately 3.9 and 4.9 days' wages when purchased from private pharmacies and online marketplaces, respectively. CGMs require spending one month's salary to purchase a CGM reader and two months' salary to buy a month's supply of CGM sensors.

**Table 2. Retail prices of self-monitoring blood glucose (SMBG) devices by sector.**

| Sector | SMBG device | Median price per unit (US$) | Minimum price per unit (US$) | Maximum price per unit (US$) |
|---|---|---|---|---|
| Public | Meter[*] | - | - | - |
| | Test strip[*] | - | - | - |
| | Lancet | 0.05 | 0.01 | 0.1 |
| Private | Meter | 23.9 | 16.15 | 44.81 |
| | Test strip | 0.34 | 0.09 | 0.63 |
| | Lancet | 0.02 | 0.01 | 0.13 |
| Online marketplace | Meter | 18.37 | 3.13 | 131.19 |
| | Test strip | 0.27 | 0.09 | 0.96 |
| | Lancet | 0.02 | 0.01 | 2.75 |
| | CGM reader | 106.70 | 64.25 | 279.19 |
| | CGM sensor | 106.26 | 68.62 | 341.59 |

SMBG = self-monitoring blood glucose, CGM = Continuous glucose monitoring

**Table 3. Affordability of self-monitoring blood glucose (SMBG) devices by sector.**

| Sector | SMBG device | Affordability (days' wages) |
|---|---|---|
| Private | Meter | 6.8 |
| | Test strips | 4.9 |
| Online marketplace | Meter | 5.3 |
| | Test strips | 3.9 |
| | CGM reader | 30.5 |
| | CGM sensor | 60.7 |

CGM = continuous glucose monitoring

## Prices and affordability of self-monitoring versus insulin treatment

Fig 4 shows the affordability of 30 days of diabetes management using analogue pre-filled pens, pen needles, meter, test strips and lancets. When purchased in private pharmacies, the cost of testing exceeded that of the analogue pens and pen needles. Conversely, testing-related costs were lower than that of analogue insulin and pen needles in online platforms. Overall, affordability of insulin and self-monitoring in private pharmacies and online marketplaces was poor with the lowest paid unskilled government worker spending nearly all their monthly salary to buy 30 days' supply of total treatment (see Fig 4). The situation in the public sector was not much better. While insulin is free-of-charge to people covered by JKN, OOP payments were needed to purchase pen needles and SMBG devices from private pharmacies or online, requiring approximately 15–19 days' wages.

## Discussion

To the best of our knowledge, this study is the first national survey to include SMBG devices (availability, prices, and affordability) in Indonesia. Very few similar studies have been

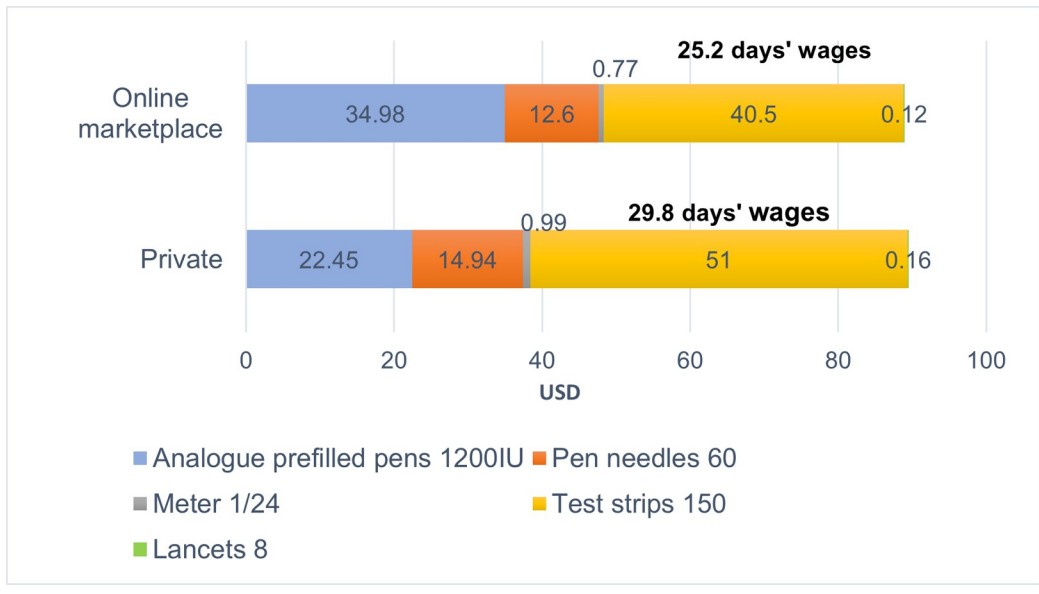

**Fig 4. Prices and affordability for 30 days' supply of total treatment (insulin, delivery device and self-monitoring blood glucose devices).**

conducted in other countries as most diabetes studies have largely focused on the availability, price and affordability of medicines.

## Availability

Availability of analogue insulins ranged from low to fairly high depending on the sector. Analogue insulins were slightly more available in public facilities (63.6%) than in private pharmacies (43.2%). These figures were below the WHO target of 80% availability of affordable essential medicines for non-communicable diseases [22]. The fact that insulin was more available in the public sector was understandable. The majority of Indonesian patients are JKN members, so they could access healthcare (including insulin) at no cost from JKN-participating facilities (mainly public hospitals). Patients can readily obtain insulin for free from private pharmacies through the referral-back scheme in which JKN-partnering private pharmacies supply insulin for JKN patients. However, a limited number of private pharmacies participate in the referral-back scheme which may hinder patients' access to insulin. Further, no human insulin was found in either sector. In the public sector, human insulin was available through the electronic catalogue, a platform for purchasing medicines and medical devices for the JKN scheme. However, human insulin was not provided to outpatients by public hospitals in our study. Similarly, a study conducted in a major hospital in Indonesia in 2016–2017 found analogues constituted 99% of insulin use, leaving an extremely small proportion of patients using human insulin. The predominant use of analogues might be due to preferences of doctors resulting in low availability of human insulin in the Indonesian market as no doctors prescribe it [23]. The Indonesian Association of Endocrinologist publishing guidelines on diabetes treatment provides no specific explanation on the selection and superiority of insulin types (analogue vs human insulin) [24].

Other studies have shown this varying availability with a multicenter study of 13 LMICs collecting insulin data in 2016 found similar availability of analogues (55%-63%) in public outlets, but lower availability of analogues in private pharmacies (27%-36%). However, that study documented good availability of human insulin in both private and public sectors. Insulin availability was also documented in a 2018 study involving 56 hospitals in a major city of China with the availability of analogues ranging between 60%-100% and human insulins 8%-100% depending on the type of insulin and the hospital level [25].

In terms of SMBG devices, our study observed their availability in the private sector was higher than the public sector. In public facilities, laboratory-based glucose testing was used more frequently than in-facility testing using a meter and strips. In studies in Kyrgyzstan, Mali, and Peru the availability of SMBGs in the private sector was higher than in the public, but the opposite was noted in Tanzania [26].

## Price

Prices of analogue insulin varied considerably across three sectors with the lowest median price found in the public sector (US$ 5.26 per/1000IU), followed by private pharmacies (US$ 11.24) and the online marketplace (US$ 28.65). Median prices of analogue insulin in our study were far lower in all sectors than a 2016 study of thirteen LMICs with prices of analogue insulin at US$ 29.39 and US$ 43.81 in public and private sectors, respectively [27]. This promising finding highlights the effectiveness of insulin procurement prices in Indonesia. However, the price cannot be solely used to determine access as the availability of analogue insulin in Indonesia, particularly in private pharmacies, was sub-optimal.

Online marketplaces had the lowest median prices of meters and test strips compared to public and private outlets, but large price variations were found. To some extent, the prices of

meters and test strips particularly from the private sector in our study correspond to those reported in four LMICs with US$ 10.7 -US$ 31.3 for a meter and US$ 0.3 –US$ 0.91 per strip, depending on the sector [26]. The cost of test strips in high-income countries like the USA was somewhat higher at around US$ 0.98 for a strip [28].

## Affordability of insulin and self-monitoring

Our study revealed that OOP payments for insulin, delivery devices and SMBG devices, were not affordable for people on low wages in Indonesia. Patients would have to work 2–3 days to purchase analogue insulin in the outlets and eight days to buy it online. Analogue insulins in our study (excluding products purchased online) were much more affordable than in several other LMICs requiring 6–8 days' wages and 7–16 days' wages in 2016 to purchase the insulin in public sector and private pharmacies, respectively [27]. For self-testing, our study showed patients could spend 5–7 days' wages to buy a meter and 4–5 days' wages a pack of 50 test strips. If purchasing a month's supply of CGM sensors, no salary would remain for basic daily needs.

To purchase 30 days' supply of treatment (insulin pens, pen needles and self-monitoring based on 5 tests per day), patients need to spend nearly all their monthly salary. Testing-related costs (58.2% of combined treatment costs) were more expensive than analogues and pen nee-dles in private pharmacies, but the reverse was true in online marketplaces. In both sectors, the largest single cost was the test strips. A study by Ogle et al involving 15 lower-income countries also highlighted that the costs of testing supplies were responsible for the largest contribution (62%) for standard diabetes care expenditure. Nonetheless, it is important to note that Ogle et al's study was conducted in people with type 1 diabetes requiring more intensive insulin reg-imens and included HbA1c testing [29]. By contrast, some studies conducted in well-resourced countries (i.e., Germany, USA) found SMBG devices accounted for less than 30% of the total costs for insulin users (type 1 and 2 DM) [28, 30].

## Global and national measures for improving access to insulin and glucose self-monitoring devices

Access to insulin, delivery devices and glucose self-monitoring devices relies on many factors including their availability and affordability. Therefore, a range of measures at global and national level should be undertaken to ensure insulin and these devices are available and affordable concomitantly.

**WHO prequalification scheme.** Currently, three multinational companies (Eli Lily, Sanofi and Novo Nordisk) control over 95% of global insulin supply [31]. Other insulin manufacturing companies have been identified, including producers in India and China, yet their impact on the global market is insignificant [32]. To address the limited technical capac-ity of regulatory agencies particularly in LMICs to undertake assessment in the dossiers of bio-logical products, WHO launched the prequalification for insulin in late 2019 [33]. This strategy for human insulin and long-acting analogue insulin may facilitate the manufacturers of insulin to submit their products for regulatory review by the WHO, thus enabling competi-tion whilst maintaining safety and efficacy of the products. In this sense, the inclusion of essen-tial insulin on the WHO's Prequalification Programme is an opportunity to facilitate entry of new companies into the market. The similar strategy has been adopted for SMBG devices with the aim of widening access in less-resourced settings [34].

**WHO model list of essential in vitro diagnostics.** WHO published the first edition of List of Essential in Vitro Diagnostics in 2018 with the latest edition in 2020. This list provides evidence-based guidance for each country to develop or update its national list of essential diagnostics. For less-resourced countries in particular, the list may benefit to prioritize the

procurement of in vitro diagnostics. The latest edition of the list includes meters for glucose testing for type 1 and type 2 diabetes in community settings and health facilities without laboratories. The list also encompass other SMBG supplies (e.g., strips, lancets) as part of the essential diagnostics for diabetes [35].

**Encouraging manufacturers to produce interoperable SMBG devices.** The market for SMBG devices was worth more than USD 24.3 billion in 2022 and is anticipated to expand at over 9.5% compound annual growth rate from 2023–2032. Globally, there are more than 100 companies marketing glucose monitoring systems with the majority located in North America and European Countries i.e., USA, Switzerland, Germany, Japan [36]. For the sake of technical and operational upgrades, new glucose meters are produced every few years along with specifically designed test strips resulting in incompatibility of new test strips with the old meter. Further, test strips are designed for specific meter models so they cannot be used across different brands or even across different models of the same brands. Interoperability has been considered as an essential part in diabetes treatment. The US Food and Drug Administration has created new regulatory classifications and introduced new designations for interoperable insulin delivery devices i.e., glucose sensors, insulin pumps and controllers. Those new classifications encourage manufacturers to design insulin delivery devices that are compatible with the existing ones [37]. Similar regulation might be applied to SMBG devices to encourage manufacturers to produce durable and interoperable SMBG devices, thus maximizing advantage to patients.

**Mark-ups regulation within the supply chain.** After procurement of products from manufacturers, prices increase within the distribution chain due to add-ons such as mark-ups (importers, wholesalers, retailers), transportation costs, taxes, dispensing fees etc. These charges have been found to add 8.7%-47.7% to the manufacturer's selling price for locally produced insulins and 10.0%-565.8% for imported insulins in some LMICs. Similarly, cumulative mark-ups for glucose monitoring products can be high ranging between 50%-200% of the manufacturers' selling price [38]. For Indonesia, the cumulative mark-up ranged within 98%-111% for JKN patients, but considerably higher i.e., 89%-566% for non-JKN patients [39]. This shows that mark-ups contribute considerably to patient prices, thus regulating mark-ups should result in lower prices and improved affordability. However, price and mark-up regulation may not necessarily lower patient prices of medicines and devices [40]. Therefore, any pricing and mark-up regulations should be incorporated with other strategies (e.g., enforcement, price monitoring, regular evaluation) to ensure their efficacy without resulting in unexpected detrimental effects (e.g., unavailability, poor quality) [38]. In the context of Indonesia, insulins and nearly all SMBG devices are imported. Indonesia has price regulations (e.g., procurement price setting, distribution margins), thus enabling decreased wholesale and retail mark-ups in public and private facilities affiliating with JKN [38]. However, the enactment of mark-ups regulation should be expanded in the private sectors as that is where people spend their OOP payments

**Inclusion of SMBG in the NHI scheme.** NHI schemes in a range of countries cover treatment for diabetes, including the management of complications, but with diverse coverage levels. However, few schemes in LMICs provide reimbursement for glucose self-monitoring devices. Tanzania (a lower-middle income country) through its NHI Fund provides free insulin, syringes and test strips for its members in public facilities. Nonetheless, availability remained a concern, especially in the private facilities. Likewise, poor availability and affordability of diabetes-related supplies were also observed in Peru, Mali and Kyrgyzstan despite the presence of NIH [26]. The inclusion of SMBG devices in the NIH may address problems related to availability and affordability, but this solution might be successfully applicable in well-resourced settings. Constrained access to SMBG devices is nearly non-existent in high-

income countries where adequate availability and affordability of meters and test strips was observed in primary care facilities [41]. In the United Kingdom, test strips are provided at no charge from the National Health Service facilities [42], whilst Australian residents pay negligible co-payments for purchasing strips [43]. With respect to Indonesia's JKN, this public insurance has provided insulin for-free for people with diabetes using insulin. Nevertheless, it should also provide meters, strips, and lancets as the part of its coverage for optimizing treatment outcomes.

**Government incentives for local manufacturers.** The Covid-19 pandemic has provided some lessons worldwide including resilience in pharmaceutical supplies. In the case of Indonesia, the government has issued some policies related to local content requirement to support locally produced medicines and medical devices. The products with high local content (minimum 50%) might obtain high priority to be purchased within public procurement [44]. In addition, efforts have been undertaken by the Indonesian government to ensure the capability to locally manufacture medical devices and equipment. Further, the government included the medical device industry as one of the priority industries in the 2015–2023 National Industrial Development Master Plan. It is interesting to note that the number of medical device manufacturers has increased considerably. In 2018–2019, Indonesia only had 200 companies. That number had multiplied several times to 1,043 industries in 2023. This strategy is expected to reduce the nation's reliance on imported medical devices to 75% by 2025 [45]. However, The Master Plan would benefit people with diabetes if it can support the establishment of companies manufacturing insulin and SMBG devices.

## Limitations and recommendations of the study

Our study has several limitations. Firstly, availability refers to the supply of the products on the day of data collection so our research may underestimate the availability as the healthcare facilities may have placed the order and awaited its delivery. However, this circumstance reflects the real-life situation experienced by the patients seeking their diabetes-related supplies. Secondly, prices were calculated based on 1000IU of insulin and 5 tests a day. This may not reflect the true costs of insulin treatment and SMBG for people paying OOP. Thirdly, the lack of a universally accepted definition of affordability may impede the accurate comparison of our findings with that of other studies [4, 17, 46]. Use of salary of lowest paid unskilled government workers may underestimate the true situation as a sizeable proportion of the population earns less than this worker [5]. The study used an adaptation of the WHO/HAI methodology. However, due to the absence of an updated international MSH drug pricing indicator guide [47], prices were not compared to an external benchmark hence a median price ratio was not calculated.

Access to self-testing devices is substantially determined by prices where people have to pay OOP. Undertaking a price component study of meters and strips may be beneficial. The study should inform the add-on costs along the supply chain, thus enabling the government and other stakeholders to undertake effective interventions to contain mark-ups and other charges resulting in lower final prices to SMBG device users. Additionally, longitudinal studies for regular monitoring of availability, price and affordability of insulin and SMBG devices will be essential to inform policies and practices to improve self-monitoring of glucose levels and hence clinical outcomes of the disease.

## Conclusion

These findings provide a snapshot on the availability, price and affordability of insulin, delivery devices and SMBG devices in Indonesia. Our study revealed availability of analogue insulin

was fairly high in public healthcare facilities, but low in private pharmacies. Conversely, better availability was observed for SMBG devices in the private sector than the public where meters and strips are not supplied for self-use. For those on low incomes, analogue insulin and SMBG devices were unaffordable regardless of the sources of the purchase. As there is no single effective measure to address problems associated with availability, price and affordability of insulin and SMBG devices, multifaceted approaches are required to improve access to affordable insulin and self-monitoring glucose devices.

## Supporting information

**S1 File.**
(XLSX)

## Acknowledgments

We would like to express our greatest appreciation to the participants of this study for agreeing to take time to answer the questions during the survey.

## Author Contributions

**Conceptualization:** Hesty Utami Ramadaniati, Yusi Anggriani, Molly Lepeska, Margaret Ewen.

**Data curation:** Hesty Utami Ramadaniati, Margaret Ewen.

**Formal analysis:** Margaret Ewen.

**Funding acquisition:** Molly Lepeska.

**Investigation:** Margaret Ewen.

**Methodology:** Hesty Utami Ramadaniati, Yusi Anggriani, Molly Lepeska, David Beran, Margaret Ewen.

**Project administration:** Molly Lepeska.

**Supervision:** Yusi Anggriani, David Beran, Margaret Ewen.

**Validation:** Hesty Utami Ramadaniati, Yusi Anggriani, Molly Lepeska, David Beran, Margaret Ewen.

**Visualization:** Margaret Ewen.

**Writing – original draft:** Hesty Utami Ramadaniati.

**Writing – review & editing:** Hesty Utami Ramadaniati, Yusi Anggriani, Molly Lepeska, David Beran, Margaret Ewen.

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
