## [Decision Letter · Decision Letter 0]

12 Jun 2024

PONE-D-24-04996Availability, price and affordability of insulin, delivery devices and self-monitoring blood glucose devices in IndonesiaPLOS ONE

Dear Dr. Ramadaniati,

Thank you for submitting your manuscript to PLOS ONE. After careful consideration, we feel that it has merit but does not fully meet PLOS ONE’s publication criteria as it currently stands. Therefore, we invite you to submit a revised version of the manuscript that addresses the points raised during the review process.

We look forward to receiving your revised manuscript.

Kind regards,

Diana Laila Ramatillah, PhD

Academic Editor

PLOS ONE

Journal Requirements:

A clean copy of the edited manuscript (uploaded as the new *manuscript* file)”.

4. We note that your Data Availability Statement is currently as follows: [All relevant data are within the manuscript and its Supporting Information files]

5. Please include a copy of Table 4 which you refer to in your text on page 12.

Additional Editor Comments:

Authors need to revise the methodology based on the reviewer comments. Authors also need to revise the discussion and the conclusion based on the aim of this study. They need to explain more detail about the findings.

Reviewers' comments:

Reviewer's Responses to Questions

**Comments to the Author**

1. Is the manuscript technically sound, and do the data support the conclusions?

Reviewer #1: Yes

Reviewer #2: Yes

2. Has the statistical analysis been performed appropriately and rigorously? 

Reviewer #1: Yes

Reviewer #2: Yes

3. Have the authors made all data underlying the findings in their manuscript fully available?

Reviewer #1: Yes

Reviewer #2: Yes

4. Is the manuscript presented in an intelligible fashion and written in standard English?

Reviewer #1: Yes

Reviewer #2: Yes

5. Review Comments to the Author

Reviewer #1: This is a very interesting study with important findings. However, the study does not exactly follow WHO/HAI methodology while doing pricing calculations. This needs to be described clearly in the methods/discussion and limitations sections of the study.

Kindly mention the following

The study does not exactly follow a WHO/HAI methodology but a variation and the prices are reported in USD (instead of a median price ratio)

This could be partly because of the absence of an updated international MSH drug pricing indicator guide (as the last one was published in 2015)

Also at places, sentences and paragraphs need support references (for example below)

Thirdly, the lack of a universally accepted definition of affordability may impede the accurate comparison of our findings with those of other studies.

Reviewer #2: 1. Line 176: The authors have used 150 test strips. Is this for a month or the entire two years? If this is meant for a month, I would suggest to use the Indonesian guidelines. Indonesia, being a middle income country, it is less likely for a diabetic to do 6-8 self-assessment per day.

2. Line 203-204: “As human insulin was not available in public health facilities, it is unsurprising that no insulin syringe was provided in this sector.” I tend to defer with the authors as to the unavailability of insulin syringes in public facilities. This is because of certain types of analogue insuring requires syringes for administration. Please re-look at this.

3. Line 396-451. This section seems to be new and does not relate to the title and the flow of the rest of the document. If the authors need to include it, they need to either alter the title, or add another secondary objective of the study to match with the discussion

6. PLOS authors have the option to publish the peer review history of their article (what does this mean?). If published, this will include your full peer review and any attached files.

Reviewer #1: **Yes: **Zaheer-Ud-Din Babar

Reviewer #2: **Yes: **Dr. Felix Khuluza

---

## [Author Response · Author response to Decision Letter 0]

11 Jul 2024

PONE-D-24-04996

Availability, price and affordability of insulin, delivery devices and self-monitoring blood glucose devices in Indonesia

Dear Academic Editor and Reviewers,

We sincerely express our gratitude for your insightful and critical comments on our manuscript. We have modified the manuscript in response to your remarks. We do hope that the revised manuscript will be considered by PLOS ONE. We will respond to the comments point counter point.

Journal Requirements:

1. When submitting your revision, we need you to address these additional requirements. Please ensure that your manuscript meets PLOS ONE's style requirements, including those for file naming.

RESPONSE: we have prepared the revised version and its supporting documents, so they are in line with PLOS ONE’s style requirements.

Upon resubmission, please provide the following: The name of the colleague or the details of the professional service that edited your manuscript

RESPONSE: we have not used professional editing services. Three of our co-authors are native English-speakers so they assisted by editing the revised manuscript. 

RESPONSE: we have provided the ‘Funding Information’ including the grant number. We ensure that the ‘Funding Information’ matches with ‘Financial Disclosure’.

4. We note that your Data Availability Statement is currently as follows: [All relevant data are within the manuscript and its Supporting Information files]

RESPONSE: The submission includes the aggregated data (minimal data set). The submission does not include facility-level data as the names of the facilities are confidential.

5. Please include a copy of Table 4 which you refer to in your text on page 12.

RESPONSE: we have checked our manuscript and realized that we made an error. There is no Table 4. It should have been written as Table 2 which we have corrected in the revised manuscript.

Additional Editor Comments:

Authors need to revise the methodology based on the reviewer comments. Authors also need to revise the discussion and the conclusion based on the aim of this study. They need to explain more detail about the findings.

RESPONSE: We have made some changes on Methods, Discussion and Conclusion based on comments from reviewers. The changes can be seen in the manuscript now revised with track changes.

Reviewers' comments:

Reviewer's Responses to Questions

Comments to the Author

Reviewer #1

1. This is a very interesting study with important findings. However, the study does not exactly follow WHO/HAI methodology while doing pricing calculations. This needs to be described clearly in the methods/discussion and limitations sections of the study.

Kindly mention the following

The study does not exactly follow a WHO/HAI methodology but a variation and the prices are reported in USD (instead of a median price ratio)

This could be partly because of the absence of an updated international MSH drug pricing indicator guide (as the last one was published in 2015)

RESPONSE: Thank you for your suggestion. We did not report price ratios as we felt no suitable external reference price (including one that lists prices for medical devices) was available. We did not report in Indonesian Rupiah as Plos One is a global journal.

We have modified the Methods by providing information on the variation (i.e. price reporting in US$ instead of median price ratios or prices in local currency) that we made from the original version of WHO/HAI methodology (manuscript with track changes line 28-29, 177-178). We have also acknowledged the variation as the part of limitations of this study (manuscript with track changes line 536-538).

2. Also at places, sentences and paragraphs need support references (for example below)

Thirdly, the lack of a universally accepted definition of affordability may impede the accurate comparison of our findings with those of other studies.

RESPONSE: we have provided the references to support some sentences (Manuscript with track changes line 534,536)

We agree the lack of a universally accepted definition of affordability is a challenge. The co-authors from HAI have lobbied WHO to develop a definition which we have been told is in process.

Reviewer #2

1. Line 176: The authors have used 150 test strips. Is this for a month or the entire two years? If this is meant for a month, I would suggest to use the Indonesian guidelines. Indonesia, being a middle-income country, it is less likely for a diabetic to do 6-8 self-assessment per day.

RESPONSE: 150 test strips are used for a month’s supply. If we refer to the Indonesian Guidelines for Diabetes Management published by Indonesian Association of Endocrinologist in 2021, the number of tests per day is not explicitly stated. However, the association recommends self-testing several times a day for uncontrolled patients including before each meal, 2 hours after each meal and before bedtime prior meals. The frequency of self-testing recommended in the Indonesian guideline is similar with that of American Diabetes Association. This is now stated in the revised manuscript. 

2. Line 203-204: “As human insulin was not available in public health facilities, it is unsurprising that no insulin syringe was provided in this sector.” I tend to defer with the authors as to the unavailability of insulin syringes in public facilities. This is because of certain types of analogue insuring requires syringes for administration. Please re-look at this.

RESPONSE: Currently analogue insulins in the Indonesian market are available only as pre-filled pens. Analogues in cartridges or vials require syringes to inject into pumps, but neither cartridges or vials were found.

3. Line 396-451. This section seems to be new and does not relate to the title and the flow of the rest of the document. If the authors need to include it, they need to either alter the title, or add another secondary objective of the study to match with the discussion

RESPONSE: We do agree with the reviewers to modify our objectives (see manuscript with track changes line 138-139)

---

## [Editor Report · Decision Letter 1]

12 Aug 2024

Availability, price and affordability of insulin, delivery devices and self-monitoring blood glucose devices in Indonesia

PONE-D-24-04996R1

Dear Dr.Ramadaniati

We’re pleased to inform you that your manuscript has been judged scientifically suitable for publication and will be formally accepted for publication once it meets all outstanding technical requirements.

Kind regards,

Diana Laila Ramatillah, PhD

Academic Editor

PLOS ONE

Additional Editor Comments (optional):

We have received all the revisions
---

## [Editor Report · Acceptance letter]

14 Aug 2024

PONE-D-24-04996R1 

PLOS ONE

Dear Dr. Ramadaniati, 

I'm pleased to inform you that your manuscript has been deemed suitable for publication in PLOS ONE. Congratulations! Your manuscript is now being handed over to our production team.

Kind regards, 

on behalf of

Prof Diana Laila Ramatillah 

Academic Editor

PLOS ONE